# The Sequential Mediating Effects of Dietary Behavior and Perceived Stress on the Relationship between Subjective Socioeconomic Status and Multicultural Adolescent Health

**DOI:** 10.3390/ijerph18073604

**Published:** 2021-03-31

**Authors:** Youlim Kim, Hyeonkyeong Lee, Mikyung Lee, Hyeyeon Lee, Sookyung Kim, Kennedy Diema Konlan

**Affiliations:** 1Mo-Im Kim Nursing Research Institute, College of Nursing and Brain Korea 21 FOUR Project, Yonsei University, Seoul 03722, Korea; goshimak@naver.com (Y.K.); lmk425@yuhs.ac (M.L.); leeleah86@gmail.com (H.L.); 2Mo-Im Kim Nursing Research Institute, College of Nursing, Yonsei University, Seoul 03722, Korea; sookyungkimm@gmail.com (S.K.); dkkonlan@uhas.edu.gh (K.D.K.)

**Keywords:** ethnic groups, minority groups, adolescent, socioeconomic factors, social determinants of health, health behavior, breakfast, stress, diagnostic self-evaluation

## Abstract

Studies have examined the impact of social determinants of health on the health behaviors and health statuses of ethnic minority adolescents. This study examines the subjective health of this population by examining the direct effects of multicultural adolescents’ subjective socioeconomic status (SES) and the sequential mediating effects of their dietary behaviors and perceived stress. We utilized secondary data of 500 middle school students from multicultural families who participated in the 15th Korean Youth Health Behavior Survey, 2019. Information about SES, perceived stress, subjective health status, and dietary behavior (measured by the breakfast intake frequency during the prior week) were utilized. For the relationship between the SES and the subjective health status, we confirmed the sequential mediating effects of breakfast frequency and perceived stress using SPSS 25.0 and PROCESS macro with bootstrapping. The results showed that SES had a direct effect on subjective health status and indirectly influenced subjective health status through the sequential mediating effect of breakfast frequency and perceived stress. However, SES had no direct effects on perceived stress. These findings emphasize that broadening the community-health lens to consider the upstream factor of SES when preparing health promotion interventions is essential to achieving health equity for vulnerable populations.

## 1. Introduction

Social determinants of health refer to the conditions necessary for living that influence health behaviors, such as adequate nutrition intake [1]. In particular, it was reported that in the case of low-income families, the socioeconomic inequity of minority adolescents resulting from differences in income, medical insurance, and education levels accumulated from childhood negatively affect their quality of life and health status as adults [2,3,4]. Adolescent health is shaped by the interaction of social determinants of health as well as risk and protective factors affecting health behavior [5]. Family socioeconomic status (SES) has long been identified as a strong determinant of adolescent health, and groups with low family SES generally exhibit lower health status, such as poor mental health and reduced wellbeing [6,7]. On the other hand, family SES tends to have less influence on health among adolescents than for other age groups [7]; hence, there is a lack of clarity regarding the relationship between family SES and adolescent health.

A multicultural family refers to a family composed of married immigrants or foreigners who have obtained nationality in the Republic of Korea (South Korea) [8]. Multicultural families in South Korea tend to be a vulnerable population, receiving various types of policy support such as education and welfare from the government, and their household income is reported to be 60% of the average monthly income of general Korean households ($2417 compared to about $4014) [9]. 

The birth rate of multicultural families accounted for 5.5% of the national birth rate in 2018, a two-fold increase from 10 years ago [10]. In 2018, the number of children in the age range of 13 to 18 years (defined as adolescents) from multicultural families increased by 21.7% from the previous year, and preschool children (those under the age of six years) were the largest group, accounting for 48.1% of the population [11]. Accordingly, it is expected that the number of multicultural adolescents who are at the developmental stage of middle adolescence (age 14–16 years) will increase rapidly. This continuously increasing trend suggests that research is needed to improve healthcare management for children and adolescents in multicultural populations.

Cognitively and behaviorally, middle adolescence is distinguished from early adolescence in that during the former, reasoning skills begin to develop, behavior-taking is considered, and a sense of identity develops [12]. Middle adolescence is considered the best time to prepare the foundation for healthy growth [13], and ensure holistic development (physical, psychosocial, and cognitive) and sustainable health practices are established through various sources of knowledge, experiences, and preparing their own healthcare management capability [14]. Adolescence represents the transition to adulthood from childhood, and because physical, mental, and physiological growth and development take place actively, the balance of nutritional intake is essential for proper growth. As the dietary habits of adolescents are directly related to health problems, such as various diseases and obesity, even after they become adults, it is necessary to pay attention so that adolescents can have a healthy diet habit [15]. In particular, having breakfast in adolescence is among the health behaviors considered to be essential because it is necessary for performing learning activities and for maintaining health status; breakfast-eating is the driving force behind everyday life [16]. Adolescents’ breakfast intake was found to be related to the economic level of households, which is one of the social determinants of health, and the rate of skipping breakfast was found to be high among low-income families [17,18]. 

According to a 2018 national public survey on ethnic minority families in Korea, the rate of skipping breakfast among adolescents from low SES families (30.1%) was about twice as high as that of the group with high SES. On further investigation of the reasons for skipping breakfast, it was found that “not prepared meals” was cited as a reason more than twice the number of times in the low SES group than in the high SES group [19]. This gap highlights the issue of health for minority adolescents with low household SES. 

Earlier studies have shown that skipping breakfast is associated with poor mental health, and more frequent breakfast skipping results in higher perceived stress among adolescents [20,21]. High stress was also found to have negative effects on adolescents’ subjective health [22]. Notably, it was found that adolescents’ perceptions of SES become more strongly linked to their subjective health [23] as adolescents are expected to become more aware of and responsive to social hierarchies [24]. However, the role of important health behaviors or statuses, such as the frequency of eating breakfast and perceived stress level, in the relationship between subjective health status and household SES has not been clearly determined. Therefore, this study examined the direct effects of subjective SES on the subjective health status of multicultural households with adolescents; we also examined the causal relationship between dietary behavior and perceived stress on subjective health status. Therefore, we proposed the following hypotheses.

**Hypothesis** **1.**
*Subjective SES would have a direct effect on subjective health status.*


**Hypothesis** **2.**
*Subjective SES would have an indirect effect on subjective health status via the mediator of breakfast frequency.*


**Hypothesis** **3.**
*Subjective SES would have an indirect effect on subjective health status via the mediator of perceived stress.*


**Hypothesis** **4.**
*Subjective SES would have an indirect effect on subjective health status via sequential mediation of breakfast frequency and perceived stress.*


## 2. Materials and Methods

### 2.1. Design

This study involves a cross-sectional secondary analysis using data from the 15th Korea Youth Risk Behavior Survey (KYRBS) of 2019 by the Korea Disease Control and Prevention Agency. It employed a descriptive research design to verify the sequential mediating effect of dietary behavior and perceived stress on the relationship between subjective SES and subjective health status of middle adolescents from multicultural families.

### 2.2. Sample

This study was approved by the Institutional Review Board of Yonsei University Health System (Y-2020-0195). This study used national public data from the 15th KYRBS. Since 2005, the survey has been conducted on middle and high school students in South Korea to determine the current status of health behaviors such as smoking, drinking, physical activity, eating, and mental health. In April 2019, the 15th Youth Health Behavior Survey was conducted with 60,100 students (response rate: 95.3%) from 800 schools in 17 cities and provinces. The cities and schools were determined through population stratification, sample distribution, and sampling. The sample comprised multicultural adolescents, defined as those whose parents’ nationalities were not South Korean; 500 of the 57,303 participants in the survey were classified as middle school students from multicultural families. Table 1 presents the sociodemographic characteristics of the participants. The mean age of the adolescents was 13.31 ± 0.99 years. Nearly half of the respondents were male (47.6%), and approximately one-half (48.0%) were from suburban areas. The majority of the adolescents’ living arrangements were family cohabitation (96.0%).

### 2.3. Variables

#### 2.3.1. Subjective Socioeconomic Status

The subjective socioeconomic status was the household’s economic status as perceived by the adolescents and was rated on a five-point scale from 1 (high) to 5 (low). The response was back-coded so that the higher the score, the higher the economic status.

#### 2.3.2. Dietary Behavior

Dietary behavior was measured by the breakfast intake frequency during the past week; it was measured from 0 to 7 days. Consuming only milk or juice was excluded as breakfast intake.

#### 2.3.3. Perceived Stress

Perceived stress was measured by the question “How much stress do you usually feel?” The scores ranged from 1 (very much) to 5 (not at all). Each question was back-coded so that the higher the score, the higher the stress.

#### 2.3.4. Subjective Health Status

Subjective health status was measured as a one-item question rating one’s usual health status on a five-point scale from 1 (very healthy) to 5 (very unhealthy). The response was back-coded so that the higher the score, the higher the health status.

#### 2.3.5. Sociodemographic Variables

We selected the individual level sociodemographic variables, including sex, residential area, and living arrangement based on previous studies [25,26], related to the adolescents’ health behavior national survey. Residential area was divided into “rural”, “suburban”, and “urban”. Living arrangement was categorized as “home with family”, “home with relatives”, “dormitory”, and “care facility”.

### 2.4. Analysis

The descriptive statistics, such as frequency, mean, and standard deviation of the participants’ sociodemographic variables and study variables (subjective SES, dietary behavior, perceived stress, and subjective health status) were analyzed using IBM SPSS Statistics 25.0 (SPSS, Chicago, IL, USA). The sequential mediating effects of the variables were confirmed using SPSS PROCESS macroVer. 3.4.1 (Model 6). The association between subjective SES, breakfast frequency, perceived stress, and subjective health status was presented as the results of the regression analysis in PROCESS macro. The significance of the indirect effects was verified with a bootstrapping procedure (95% CI, k = 10,000) [27]. This procedure estimates the standard error of indirect effects through simulation; if the confidence interval does not include zero, there was a statistically significant indirect effect. All direct and indirect effects were reported using the unstandardized regression coefficient (B) [27].

## 3. Results

### 3.1. Descriptive Statistics and Correlations of Key Variables

Subjective SES was described as high (8.4%), upper-middle (22.2%), middle (51.8%), lower-middle (14.2%), or low (3.4%). Just under half of the participants had a breakfast frequency of three days or less (43.6%), and a fifth of the adolescents (20.4%) did not consume breakfast every day of the week. The score of perceived stress was 3.21 ± 1.01, and subjective health status was 3.82 ± 0.89. 

The results of the correlation analysis are shown in Table 2. There was a positive correlation between the subjective SES scores and the frequency of breakfast consumption and subjective health status, but it was not related to perceived stress. Adolescents’ breakfast frequency was negatively related to perceived stress and positively related to subjective health status. Moreover, perceived stress was negatively related to subjective health status. 

### 3.2. Association between Subjective SES, Breakfast Frequency, Perceived Stress, and Subjective Health Status

The results of the regression analysis for each pathway are presented in Table 3 and Figure 1. The pathway coefficient for the path from subjective SES to breakfast frequency was positive and significant. Subjective SES was not a significant predictor of perceived stress but was a significant negative predictor of breakfast frequency. Meanwhile, subjective SES and breakfast frequency were significant positive predictors of subjective health status. The path coefficient for the path from perceived stress to subjective health status was negative and significant. 

### 3.3. Sequential Mediating Effects Among Variables

The bootstrapping analysis of the sequential mediating effects (Table 4) reveals the following: first, breakfast frequency had a significant mediating effect on the relationship between the subjective SES and subjective health status (95% CI 0.0002–0.0271); second, perceived stress did not have a mediating effect on the relationship between the subjective SES and subjective health status of adolescents (95% CI −0.0193 to 0.0373); third, subjective SES had an indirect effect on the subjective health status with the sequential mediating effect of breakfast frequency and perceived stress (95% CI 0.0002–0.0098); and fourth, the total effect of subjective SES on subjective health status was significant (B = 0.201, 95% CI 0.1160–0.2864). Even after breakfast frequency and perceived stress were included as mediators, subjective SES still had a significant direct effect on subjective health status (B = 0.178, 95% CI 0.0974–0.2590). Therefore, the partial mediating model was confirmed. In other words, for multicultural adolescents, a higher subjective SES level is associated with higher breakfast frequencies, lower levels of perceived stress, and higher levels of subjective health status. 

## 4. Discussion

This study provides evidence to support the role of subjective SES of multicultural adolescents on their subjective health status and the indirect effect of mediating health behaviors, using a representative sample from national-level public data (15th KYRBS).

Previous studies focusing on individual determinants of health [28] have confirmed that the relationships of breakfast skipping, stress, depression, and health behaviors affect the quality of life for adolescents. However, this study attempted to confirm the role of the structural social determinants of health (SES) via behavioral (breakfast frequency) and psychosocial factors (perceived stress). 

Based on these results, it is meaningful to emphasize the importance of the strategies, considering breakfast frequency and perceived stress level, needed when it comes to raising the subjective health status of multicultural adolescents considering subjective SES.

### 4.1. Subjective SES and Health Behavior

In a previous study, an indicator of the household’s economic level as perceived by adolescents was whether or not they received support for meal costs; it was found that students who received support for meals costs due to their low household economic level had higher stress and depressive symptoms than students without support [29]. Another study [30], showed that the SES variables rated by adolescents had a significant effect on their own physical and mental health problems. These findings are in line with our results as they indicate the existence of health inequity based on SES, even in adolescence. 

We found support for the indirect effects of subjective SES on subjective health status by mediating perceived stress and breakfast intake, and although it is a small effect, it is meaningful evidence for the lower SES group. According to a study [31] that used data from the Korean National Health and Nutrition Examination Survey to confirm the mediating effect of dietary habits in the relationship between SES and children’s dietary quality, eating breakfast was a significant mediator in the lower-middle household income group. As dietary quality is known to be a major factor affecting subjective health status [32,33], this evidence supports our findings. However, similar to the study aforementioned [31], a very small effect on the relationship between subjective SES and subjective health status was observed in this study, indicating limited practical applications.  We cannot entirely rule out the possibility of residual confounding by other factors. Given the small magnitude of the observed effect, caution should be applied to interpret the results. 

Most national-level public databases and studies include a single SES index as a variable for family SES [34]. A previous study revealed low concurrent validity between traditional SES, the Family Affluence Scale, and subjective SES indicators in the adolescent population; hence, it is necessary to consider different approaches for measuring SES among adolescents [35]. In this regard, the subjective SES measure needs to account for various factors that reflect the current times, as individuals’ subjective perceptions regarding possession of resources are also not consistent. 

### 4.2. The Relationship between Breakfast Intake and Perceived Stress

Reduction in breakfast skipping is one of the important health management tasks as it has been selected as an indicator of the health goal of the South Korea National Health Promotion Plan 2030 [36]. Previously, most studies from extensive reviews have established the relationship between skipping breakfast and obesity [37] or academic performance [38]. In the present study of the minority adolescent population, however, the new information about a positive effect of breakfast intake on stress reduction—a psychosocial factor, based on the existing knowledge [28,39], would have substantial implications for healthcare professionals and policy stakeholders. In this study, in the group with the lowest SES, breakfast skipping on all seven days in a week was reported by 52.9% of multicultural adolescents, which was more than twice as high as that reported by non-multicultural adolescents (25.4%). Therefore, future interventions for multicultural adolescents and families to be aware of the benefits of having breakfast are suggested, and the interventions need to be prioritized with those who are socioeconomically vulnerable.

### 4.3. Methodological Implications of Sequential Mediating Effects

This study showed that the subjective SES rated by multicultural adolescents indirectly influenced the subjective health status by sequentially mediating health behaviors such as breakfast intake and perceived stress. Based on a previous study [40], it can be estimated that health behavior is highly likely to play a mediating role in the relationship between subjective SES and subjective health status. However, until now, most studies have not examined the effects of the mediating variables by comprehensively considering them in one model but have individually verified the mediating effects or set each mediating variable in parallel. This study uses a multiple sequential mediating model to examine the sequential mediated pathways leading to perceived stress reduction by breakfast intake frequency in the relationship between subjective SES and subjective health status, which provides the advantage of being able to simultaneously verify the influence of the predictive variables on each variable as well as the influence between each variable [41].

When each mediating effect is tested with a simple mediating model, the individual models can lead to biased mediating estimates due to the omitted variables [42]. Confirming the sequential multiple mediating effects is meaningful for verifying the overall mediating effect of the relationship between the predicted and reference variables and simultaneously verifying the mediating effect of other variables (when present). Therefore, the sequential mediating effect verification is an appropriate method for the analysis; it more organically reflects the relationship between various variables, such as health behaviors and psychosocial factors, within the social structural environment that influences the subjective health status of multicultural adolescents.

### 4.4. Limitations and Recommendations for Further Research

A few limitations in this study are evident. First, the index of the SES perceived by multicultural adolescents is a single item that represents the household’s economic level, and there is a limitation in reflecting the sociocultural and environmental characteristics of adolescents (family possessions, social capital, institutions, and policies). Future studies should move beyond a single question about economic indicators and include composite factors including parents’ occupation, educational level, parental income level, support for vulnerable groups, sociocultural indicators such as the frequency of participation in leisure activities, and the subjective social status of adolescents (i.e., adolescents’ social position within their school and peer group) [43], which need to be represented by an integrated score. Second, due to data limitations, interpersonal and community determinants (i.e., parental rearing attitude and poverty level in the residential area) could not be included among the social determinants of adolescents. If these measurements were analyzed as moderating factors, more important implications could have been determined. 

Based on the findings of this study, the following future research is recommended. This study focused on the effects of subjective SES on adolescents’ stress and health behaviors such as breakfast intake. Since relative SES may be regarded as a personal or subjective factor, it is suggested to combine an objective index for measuring SES. It should also be noted that psychosocial factors are formed by individual cognitive and perceived processes; thus, efforts to reduce socioeconomic health inequity should not be resolved only by individual care or treatment but through the integration of social, environmental, and policy alternatives. Additionally, since this is a cross-sectional study, a clear analysis of the causal relationship is necessary through longitudinal studies between subjective composite SES and health behavior and subjective health status. Additionally, health status according to SES may differ according to interaction with gender. In previous studies, dietary habits [44], stress [45], and subjective health [46] had different levels according to gender, and future studies could identify the mechanisms behind gender-specific determinants.

Meanwhile, based on these findings, the health status of low-income multicultural adolescents in South Korea is expected to become more vulnerable after the coronavirus disease (COVID-19) pandemic, further widening the health gap with non-multicultural adolescents. Due to the pandemic, prolonged school closures and parental unemployment have been reported as threats to adolescents’ health [47]. The number of children skipping meals is increasing due to the prolonged COVID-19 pandemic [48], especially among adolescents of minority ethnic families, who constituted a high proportion [49]. To address this health inequity, an institutional foundation is needed to ensure universal health coverage from a holistic perspective (considering the whole life process), including the socioeconomic environment of vulnerable children and adolescents. It is also necessary to develop health promotion policies and programs to alleviate the fundamental social and economic inequity faced by vulnerable groups such as low-income multicultural adolescents.

## 5. Conclusions

These findings highlight the importance of understanding that the SES of multicultural adolescents affects their perception of health status and health behavior. The results also emphasize that broadening the scope of interventions to include the upstream factor of SES is essential for vulnerable populations, such as multicultural adolescents and their families, to achieve health equity. In terms of practical implications, future research may explore the possibility of differences due to gender in the mediating path to clarify further the relationships between SES, dietary behavior, and subjective health status.

## Figures and Tables

**Figure 1 ijerph-18-03604-f001:**
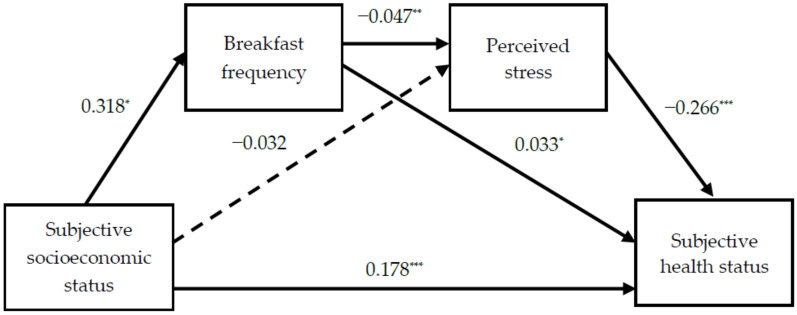
The pathways and coefficient. Solid lines represent significant pathways. The dotted line represents a non-significant pathway. * *p* < 0.05, ** *p* < 0.01, *** *p* < 0.001.

**Table 1 ijerph-18-03604-t001:** Sociodemographic characteristics of the participants (*N* = 500).

Variables	Mean ± SD or *n* (%)
Age	13.31 ± 0.99
Grade	
1st (14 years old)	206 (41.2)
2nd (15 years old)	168 (33.6)
3rd (16 years old)	126 (25.2)
Sex	
Male	238 (47.6)
Residential area	
Rural	158 (31.6)
Suburban	240 (48.0)
Urban	102 (20.4)
Living arrangement	
Home with family	480 (96.0)
Home with relatives	3 (0.6)
Dormitory	8 (1.6)
Care facility	9 (1.8)

**Table 2 ijerph-18-03604-t002:** Correlations between variables.

	Subjective Socioeconomic Status	Breakfast Frequency	Perceived Stress	Subjective Health Status
Subjective socioeconomic status	1			
Breakfast frequency	0.104 *	1		
Perceived stress	−0.042	−0.131 **	1	
Subjective health status	0.204 **	0.161 **	−0.323 **	1

Note. * *p* < 0.05, ** *p* < 0.01.

**Table 3 ijerph-18-03604-t003:** Verification of the association between breakfast frequency and perceived stress in subjective socioeconomic status (SES) and subjective health status.

Pathway			B	SE	t	*p*	F
Subjective socioeconomic status	→	Breakfast frequency	0.318	0.104	2.328	0.020	5.419
Subjective socioeconomic status	→	Perceived stress	−0.032	−0.029	−0.638	0.524	4.558
Breakfast frequency			−0.047	−0.128	−2.869	0.004	
Subjective socioeconomic status	→	Subjective health status	0.178	0.180	4.332	<0.001	29.400
Breakfast frequency			0.033	0.103	2.446	0.015	
Perceived stress			−0.266	−0.302	−7.242	<0.001	

Note: SE = standard error. Unstandardized coefficients (B) are presented.

**Table 4 ijerph-18-03604-t004:** The sequential mediating effects among variables by bootstrapping.

Mediating Pathway	Bootstrapping Estimate		95% CI	
	B	SE	LLCI	ULCI
Subjective socioeconomic status ⇒ Breakfast frequency ⇒ Subjective health status	0.011	0.0070	0.0002	0.0271
Subjective socioeconomic status ⇒ Perceived stress ⇒ Subjective health status	0.009	0.0143	−0.0193	0.0373
Subjective socioeconomic status ⇒ Breakfast frequency ⇒ Perceived stress ⇒ Subjective health status	0.004	0.0025	0.0002	0.0098

Note: CI = confidence interval; LLCI = lower limit confidence interval; ULCI = upper limit confidence interval; SE = standard error. Unstandardized coefficients (B) are presented.

## Data Availability

The data that support the findings of this study are available from the corresponding author, upon reasonable request.

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
