# Peer review of "The Sequential Mediating Effects of Dietary Behavior and Perceived Stress on the Relationship between Subjective Socioeconomic Status and Multicultural Adolescent Health"

_ijerph, 2021, doi:10.3390/ijerph18073604_

Round 1

Reviewer 1 Report

This paper reports on a cross-sectional analysis of the effect of multicultural adolescents’ subjective health perception on their subjective health status, and whether this effect is mediated by breakfast frequency and perceived stress. Overall, I think the paper has potential to be of interest to the readership of the journal. But first some changes need to be made to clarify and improve the presentation of the study. I have a number of comments listed below.

“Middle adolescence is considered the best time to prepare the foundation for healthy growth…” is there is proposed reason for this? Why not childhood?

“Adolescents’ breakfast intake was found to be related to the economic level of households, which is one of the determinants of social health…” Is economic level not one of the social determinants of health, rather than a determinant of social health? The wording is inconsistent here with the first paragraph.

“that skipping breakfast is associated with mental health” do you mean associated with poor mental health?

“Notably, youth’s perceptions of their SES reflects their health status more sensitively than other measures when they are at the age when self-concept is established.” I found this sentence hard to follow and understand, can you reword it to make the point clearer?

I think you need to grant some time to defining what subjective health status is in the introduction.

You could end the introduction by outlining some specific hypotheses surrounding the effects you were expecting to find. Perhaps even presenting these in a path diagram would be useful (especially as you later talk about the significance of specific paths in the results section)?

In section 2.2 when discussing the sample, could you give some further details surrounding the demographics of the 500 adolescents who you included in your analysis? I.e. what percentage were male/female, what was the age range… [Ok, I have just read section 3.1, I would be tempted to include this information in section 2.2 rather than within the results because it does not relate to any of the hypotheses].

“The score values of SES, dietary behaviour….” It is not clear what ‘score values’ refers to?

Can you add some more information about the bootstrapping procedure, what % confidence intervals were used? How many samples?

Can you tell us the ages in years rather than grades? International readers will not automatically know what age range 1st grade corresponds to in South Korea.

Table 1 needs a bit of reformatting so that the heading from ‘living arrangement’ onwards are in line with the first, rather than second, column.

Table 1 repeats a lot of information that is already written in sentences in section 3.1. No need to include the information in the table and text, just include it once. The same point applies for Table 2 and section 3.2, the table repeats what you have written already.

At the bottom of Table 3 you have written “1Unstandardized betas and SEs are presented. 2 Standardized betas are presented.” I could not find the superscript values within the table so this was confusing. Do you mean that Ba is the unstandardized beta and βb is the standardized beta? This need clarifying.

Section 3.3 starts with “The results of regression analysis…” but regression is not mentioned in the analysis plan (section 2.4). Just make sure that your plan matches up with what you actually do so that the reader is not confused.

Again, no need to include the B values in the text in section 3 when they are already in both Table 3 and Figure 1.

In Figure 1 you have written “Direct effect (including mediator) .178***”. Is the direct effect not the effect without the mediator? If both the direct and indirect (mediator) effects are included then wouldn’t this instead represent the total effect?

“breakfast frequency had a significant mediating effect on the relationship between the subjective SES and subjective health status (B = .011, CI .0002 to .0271).” True, but you might want to acknowledge that the effect is very small! Same goes for the sequential mediating effect.

“economic difficulties experienced in childhood are known to have an especially large and wide range of perceived impacts…” Statements like this are not very useful unless you give a couple of examples of the perceived impacts.

I think that the discussion could benefit from a bit of restructuring. For example, I would have a sub-section specifically for discussing the limitations (e.g. single item measures) rather than dispersing these throughout.

In the conclusion I found that some of you wording did not reflect the actual constructs that you had studied. E.g. you did not look at actual SES but rather perceived SES. “Perceived stress management” also did not really appear in the manuscript until now.

“the possibility of differences by subgroups…” what subgroups and why would they be interesting to look at? You did not mention sub-group analysis in your recommendations for further research section so I was surprised when it was suddenly mentioned in the conclusion.

Author Response

Reviewer’s comments

Response

“Middle adolescence is considered the best time to prepare the foundation for healthy growth…” is there is proposed reason for this? Why not childhood?

We changed this as follows: “Cognitively and behaviorally, middle adolescence is distinguished from early adolescence in that during the former, reasoning skills begin to develop, behavior-taking is considered, and a sense of identity develops.”

“Adolescents’ breakfast intake was found to be related to the economic level of households, which is one of the determinants of social health…” Is economic level not one of the social determinants of health, rather than a determinant of social health? The wording is inconsistent here with the first paragraph.

The economic level of households is among the “social determinants of health.” Thus, we corrected the phrases from “determinants of social health” to “social determinants of health.”

“that skipping breakfast is associated with mental health” do you mean associated with poor mental health?

We revised “mental health” to “poor mental health” in line with references 20 and 21.

“Notably, youth’s perceptions of their SES reflects their health status more sensitively than other measures when they are at the age when self-concept is established.” I found this sentence hard to follow and understand, can you reword it to make the point clearer?

According to developmental theories, subjective social status becomes more strongly linked to well-being/health across adolescence as adolescents are expected to become more aware and responsive to social hierarchies. Thus, we rephrased the sentence as follows:

“Notably, it was found that adolescents’ perceptions of SES become more strongly linked to their subjective health as adolescents are expected to become more aware and responsive to social hierarchies.”

You could end the introduction by outlining some specific hypotheses surrounding the effects you were expecting to find. Perhaps even presenting these in a path diagram would be useful (especially as you later talk about the significance of specific paths in the results section)?

We proposed the following research hypotheses:

Hypothesis 1. Subjective SES would have a direct effect on subjective health status.

Hypothesis 2. Subjective SES would have an indirect effect on subjective health status via the mediator of breakfast frequency.

Hypothesis 3. Subjective SES would have an indirect effect on subjective health status via the mediator of perceived stress.

Hypothesis 4. Subjective SES would have an indirect effect on subjective health status via sequential mediation of breakfast frequency and perceived stress.

In section 2.2 when discussing the sample, could you give some further details surrounding the demographics of the 500 adolescents who you included in your analysis? I.e. what percentage were male/female, what was the age range… [Ok, I have just read section 3.1, I would be tempted to include this information in section 2.2 rather than within the results because it does not relate to any of the hypotheses].

We moved the contents of the general characteristics to the 2.2 Sample section and integrated the contents of 3.1 and 3.2 of the results into one paragraph.

“The score values of SES, dietary behaviour….” It is not clear what ‘score values’ refers to?

We meant the mean score of SES and corrected it.

Can you add some more information about the bootstrapping procedure, what % confidence intervals were used? How many samples?

We added information about the bootstrapping procedure in the analysis section.

The significance of the indirect effects was verified with a bootstrapping procedure (95% CI, k = 10,000).

Can you tell us the ages in years rather than grades? International readers will not automatically know what age range 1st grade corresponds to in South Korea.

We added to this and explained the age next to the grade level as follows: “1st (14 years old).”

Table 1 needs a bit of reformatting so that the heading from ‘living arrangement’ onwards are in line with the first, rather than second, column.

Table 1 was reformatted so that the headings from “living arrangement” onwards align with the first column rather than the second column.

Table 1 repeats a lot of information that is already written in sentences in section 3.1. No need to include the information in the table and text, just include it once. The same point applies for Table 2 and section 3.2, the table repeats what you have written already.

We have summarized and rewritten the contents of Tables 1,2 and 3.

At the bottom of Table 3 you have written “1Unstandardized betas and SEs are presented. 2 Standardized betas are presented.” I could not find the superscript values within the table so this was confusing. Do you mean that Ba is the unstandardized beta and βb is the standardized beta? This need clarifying.

We modified the annotations presented in Table 3 by referring to the tables in other articles by IJERPH and only presented the unstandardized regression coefficient (B) to avoid confusion.

Section 3.3 starts with “The results of regression analysis…” but regression is not mentioned in the analysis plan (section 2.4). Just make sure that your plan matches up with what you actually do so that the reader is not confused.

To confirm the effect of the independent variable on the mediated variables and dependent variable, the regression results in the Process macro results were presented in Table 3.

We described the contents of the regression analysis in the analysis section.

Again, no need to include the B values in the text in section 3 when they are already in both Table 3 and Figure 1.

As the coefficients are described in Table 3 and Figure 1, we have removed these from the paragraph.

In Figure 1 you have written “Direct effect (including mediator) .178***”. Is the direct effect not the effect without the mediator? If both the direct and indirect (mediator) effects are included then wouldn’t this instead represent the total effect?

The direct effect means that the independent variable influences the dependent variable without going through the mediator despite the existence of a significant mediator. The phrase written in this study seems to confuse the reader. Therefore, the phrase “including mediator” was deleted.

breakfast frequency had a significant mediating effect on the relationship between the subjective SES and subjective health status (B = .011, CI .0002 to .0271).” True, but you might want to acknowledge that the effect is very small! Same goes for the sequential mediating effect.

Although the small effect from this study is similar to another study that used another national survey, a small effect size indicates limited practical application. We acknowledged this as a study limitation as follows: 

“A very small effect on the relationship between the subjective SES and subjective health status was observed in this study, indicating limited practical applications. We cannot entirely rule out the possibility of residual confounding by other factors. Given the small magnitude of the observed effect, caution should be applied in the interpretation of the results.”

“economic difficulties experienced in childhood are known to have an especially large and wide range of perceived impacts…” Statements like this are not very useful unless you give a couple of examples of the perceived impacts.

We deleted the sentence.

I think that the discussion could benefit from a bit of restructuring. For example, I would have a sub-section specifically for discussing the limitations (e.g. single item measures) rather than dispersing these throughout.

We restructured the paragraph that represents the limitations of our research in section 4.4.

In the conclusion I found that some of you wording did not reflect the actual constructs that you had studied. E.g. you did not look at actual SES but rather perceived SES. “Perceived stress management” also did not really appear in the manuscript until now.

We deleted words that were irrelevant to the results from the conclusion.

“the possibility of differences by subgroups…” what subgroups and why would they be interesting to look at? You did not mention sub-group analysis in your recommendations for further research section so I was surprised when it was suddenly mentioned in the conclusion.

We revised the sentences from “by subgroups” to “by gender.”

To present this in the conclusion, gender differences were mentioned in the discussion.

Reviewer 2 Report

The authors investigated the relationship between subjective SES and adolescent health using well designed method including bootstrapping procedure. The subject of this paper is of potential significance.

I have a few comments for the authors to address.

  1. It would be better to clarify how and why variables selected as sociodemographic status were selected based on the evidence including previous studies.
  2. It would be nice to highlight differences between this research and the previous research and the strengths of this research.

Author Response

Point 1. It would be better to clarify how and why variables selected as sociodemographic status were selected based on the evidence including previous studies.

Response 1. We selected individual-level sociodemographic variables, including geographic region, based on previous studies related to the adolescents’ health behavior national survey (line 143-145).

Point 2. It would be nice to highlight differences between this research and the previous research and the strengths of this research.

Response 2. We added more on the significance of this study at the beginning of the discussion (line 217-221):

"Previous studies focusing on individual determinants of health have confirmed that the relationships of breakfast skipping, stress, depression, and health behaviors affect the quality of life of adolescents. However, this study attempted to confirm the role of the structural social determinants of health (SES) via behavioral (breakfast frequency) and psychosocial factors (perceived stress)."

Reviewer 3 Report

This is a well-written manuscript with only a few points to be raised.

Firstly, it does need to be reviewed by a qualified statistician.

Table 1 needs extended to include all data on which the calculations are based. Also, the formatting needs some minor attention. Some of the variable headings which have subcategories have been indented. For example, Residential Area, should not be indented only rural, suburban and urban.

The Breakfast days are given as more than and less than 3 days – there must also be an equal to 3 days. These also need to be more fully included in Table 1.

In Table 3, I cannot see any of the figures in bold as per note below the table – should there be?

Otherwise a well-written paper but statistical accuracy to be checked.

Author Response

Point 1. Table 1 needs extended to include all data on which the calculations are based. Also, the formatting needs some minor attention. Some of the variable headings which have subcategories have been indented. For example, Residential Area, should not be indented only rural, suburban and urban.

Response 1. Table 1 was reformatted so that the headings from “living arrangement” onwards align with the first column rather than the second column.

Point 2. The Breakfast days are given as more than and less than 3 days – there must also be an equal to 3 days. These also need to be more fully included in Table 1.

Response 2. We modified Table 1 to present only the sociodemographic characteristics. The description of the breakfast frequency is presented as follows (line 166). Just under half of the participants had a breakfast frequency of three days or less (43.6%).

Point 3. In Table 3, I cannot see any of the figures in bold as per note below the table – should there be? Otherwise a well-written paper but statistical accuracy to be checked.

Response 3. We deleted the annotations related to the bold text.
